# Electrorefining Process of the Non-Commercial Copper Anodes

**Radmila Markovic** [1,*], **Vesna Krstic** [1], **Bernd Friedrich** [2], **Srecko Stopic** [2], **Jasmina Stevanovic** [3], **Zoran Stevanovic** [1] **and Vesna Marjanovic** [1]

1 Mining and Metallurgy Institute Bor, Zeleni Bulevar 35, 19210 Bor, Serbia; vesna.krstic@irmbor.co.rs (V.K.); zoran.stevanovic@irmbor.co.rs (Z.S.); vesna.marjanovic@irmbor.co.rs (V.M.)

2 IME Process Metallurgy and Metal Recycling, Intzestraße 3, 52056 Aachen, Germany; bfriedrich@ime-aachen.de (B.F.); sstopic@metallurgie.rwth-aachen.de (S.S.)

3 Institute of Chemistry, Technology and Metallurgy, National Institute of the Republic of Serbia, University of Belgrade, Njegoševa 12, 11000 Belgrade, Serbia; jaca@tmf.bg.ac.rs

* Correspondence: radmila.markovic@irmbor.co.rs; Tel.: +381-62-450762

**Abstract:** The electrorefining process of the non-commercial Cu anodes was tested on the enlarged laboratory equipment over 72 h. Cu anodes with Ni content of 5 or 10 wt.% and total content of Pb, Sn, and Sb of about 1.5 wt.% were used for the tests. The real waste solution of sulfuric acid character was a working electrolyte of different temperatures (T1 = 63 ± 2 °C and T2 = 73 ± 2 °C). The current density of 250 A/m$^2$ was the same as in the commercial process. Tests were confirmed that those anodes can be used in the commercial copper electrorefining process based on the fact that the elements from anodes were dissolved, the total anode passivation did not occur, and copper is deposited onto cathodes. The masses of cathode deposits confirmed that the Cu ions from the electrolyte were also deposited onto cathodes. The concentration of Cu, As, and Sb ions in the electrolyte was decreased. At the same time, the concentration of Ni ions was increased by a maximum of up to 129.27 wt.%. The major crystalline phases in the obtained anode slime, detected by the X-ray diffraction analyses, were $PbSO_4$, $Cu_3As$, $SbAsO_4$, $Cu_2O$, $As_2O_3$, $PbO$, $SnO$, and $Sb_2O_3$.

**Keywords:** electrorefining; non-commercial copper anode; waste solution; high content; Ni; Pb; Sn; Sb; passivation; anode slime





## 1. Introduction

Electrorefining of the anode materials is a purification process in order to remove the ingredients that may have a negative impact on the physical, chemical, and mechanical properties of the base materials. Almost all metals can be purified by this process but based on the production data of approximately 24 Mt in 2019, copper exceeds all other metals [1,2]. Electrolytes and other process parameters must be selected so that the anode dissolution and metal deposition take place with a high degree of efficiency and a transfer degree of ingredients from anode onto cathode is minimized. Additionally, the process parameters should prevent the passivation of the anode and enable the precipitation of appropriate physical and chemical characteristics. The values of characteristic technological parameters for the electrorefining process of some metals are different for different metals (copper, nickel, cobalt, lead, and tin) [3]. If it is necessary, the additives are added to the electrolyte to ensure a proper operation of both electrodes. The addition of thiourea, gelatin, and chlorine ions during the electrorefining process had a positive effect on minimizing the nodules', porosities', and dendrites' appearance [4]. Deposited copper surface roughness was decreased with the increase in the concentration of thiourea and gelatin [5].

Anodic passivation is one of the basic problems that occur in the industrial electrorefining process and could be explained as a copper sulfate layer formation on an anode surface [6]. $Cu_2O$ form was registered in the anodes with oxygen content in the range of 0.1–0.3%. This form was also registered in the slime present on the anode surface. Through the reaction of copper (I) oxide form and sulfuric acid, a partial dissolution of copper

oxide form occurs, whereby a salt of copper sulfate and elemental copper is formed. As the chemical dissolution of this form is more intense than the electrochemical one, the copper concentration in the electrolyte increases, which is a characteristic of the commercial process of anode copper electrorefining [7].

Raw materials for smelting are becoming more and more complex so that the effect of ingredients on the anode copper electrorefining process is constantly changing. Anodes obtained from the secondary raw materials are generally rich in nickel, lead, antimony, and tin, and a low content of selenium, tellurium, and silver has been reported [8]. During solidification, Ni is enriched in the solid phase, but some elements pass into the solid solution in copper crystals (Sb, Sn, Pb, As, and Bi) and in Cu-Pb-As-Sb-Bi oxide phases [9]. The so-called "mineralogy" of the anode directly affects the passivity of the anode, formation of suspended slime, deposition of slime, as well as the possibility of separating the useful and high-value components from the obtained slime [10]. The amount of inclusion phases, primarily oxides, is a direct consequence of the content of elements in the anode. Only $Cu_2O$ and NiO are formed during the primary crystallization, while the other forms are formed during the secondary crystallization, causing a local accumulation.

The behavior of ingredients and phases during the electrolytic refining of anodes is in principle the same as for anodes obtained from the primary raw materials, but the higher content of ingredients can cause some problems during the electrorefining process such as the increase in a cell voltage or even anode passivation [11]. The mostly inhomogeneous distribution of phases leads to a different dissolution percentage in a contact with the working electrolyte.

During the electrorefining process, the ingredients from anodes could be distributed into three groups. The first group consists of ingredients with a dissolution potential that is much more negative than a dissolution potential of copper: Ni, Fe, Zn, and Co. During the process, those elements are accumulated in the electrolyte. In the nickel-rich copper anodes, the following oxide phases can be formed: NiO, Cu-Sb-Ni, and Cu-Sn-Ni [12]. Anodes obtained from the copper-based secondary materials are characterized by the nickel content of more than 0.3% wt.%. A negative impact of Ni on the refining process is reflected in the fact that NiO accelerates the process of passivation of the anode, reduces the solubility of Cu from the anode, and generates a large amount of nickel in the anode slime [13]. The second group consists of ingredients with a dissolution potential that is much more positive than the copper dissolution potential, which prevents their dissolution in the electrolyte, but they directly pass into the anode slime: Au, Ag, Pt, Se, and Te. This group also includes the insoluble salts such as $PbSO_4$ and $Sn(OH)_2SO_4$, which pass from electrolyte into anode slime. The various Pb oxide inclusions are the major Pb carriers in anode material. Pb dissolves from anodes along with Cu and immediately precipitates as the $PbSO_4$ insoluble salt. In a copper anode, rich in Ni, Sb, and Sn content, Sn could be found in different forms of Cu-Sn-Ni oxide, such as $Cu_2NiSnO_5$ and Cu-Sb-Ni oxide, as the "kupferglimmer" form $Cu_3Ni_2$-$xSbO_6$-x where x is in the range from 0.1 to 0.2. During the electrolysis process, Sn reacts with electrolyte and precipitates in anode slime [14]. The third group consists of ingredients with a dissolution potential close to a dissolution potential of copper: As, Sb, and Bi. They are dissolved during the anode refining, and under certain conditions can be precipitated (high concentration of these elements, low concentration of Cu). Arsenic ($AsO^{-3}_4$) can also react with Sb and Bi to form an insoluble compound of As-Sb-Bi known as the "floating slime" with the predicted composition of $BiAsO_4$, $SbAsO_4$, $Sb_2O_3$, and $Bi_2O_3$ [15]. Based on the data of the Jafari et al. [15], the predicted composition of "floating slime" is $BiAsO_4$, $SbAsO_4$, $Sb_2O_3$, and $Bi_2O_3$. Additionally, the operational As/(Sb + Bi) molar ratios could be maintained at or above 2 aim to reduce the presence of floating slimes, but this ratio is lower than 2 in about 26% of electrolysis plants [16].

The aim of this work was to investigate the electrorefining process of copper anodes with increased Ni, Pb, Sn, and Sb content in comparison with the content of those elements in the commercial copper anode electrorefining process. The two types of copper anodes are proposed for testing regarding the Ni, Pb, Sn, and Sb content as the ingredients. Copper

content was also different for different anode types. Real waste sulfate solution was used as a working electrolyte. Furthermore, the effect of two different working electrolyte temperatures, where one was the same as in the commercial copper electrorefining process (T1 = 63 ± 2 °C) and the second was 10 °C higher (T2 = 73 ± 2 °C), was tested. The anode elements dissolving during each test over the duration of 72 h was defined on the basis of values of a cell voltage. The mass of dissolved anodes and the mass of anode slime were used for calculating the mass percentage of anode slime in accordance with the mass of dissolved anodes. Each calculation was performed for different working conditions. Electrolyte composition is checked on each 24 h for the main elements. The phases in anode slime samples, obtained for different experimental conditions, are defined by the X-ray diffraction (XRD) analysis.

## 2. Materials and Methods

### 2.1. Materials

#### 2.1.1. Non-Commercial Copper Anodes

Two types of non-commercial copper anodes regarding the chemical content of Ni, Pb, Sn, and Sb are prepared for the electrorefining tests. Oxygen content was under 200 ppm. The main difference between the anodes was in Ni and Cu content. In the anodes with sample code An-1, Ni content was approximately 5 wt.%, and in the anodes with sample code An-2, Ni content was approximately 10 wt.%. The total content of Pb, Sn, and Sb for both types of anodes was approximately 1.5 wt.%. The content of each element (Pb, Sn, and Sb) was about 0.5 wt.%. Copper content is calculated on the basis of values for the content of Ni, Pb, Sn, and Sb as a difference up to 100 wt.%. Cathode copper and pure metals (nickel, lead, tin, and antimony) were used for the non-commercial copper anode preparation. Preparation of a suitable mixture for melting was made in an induction furnace. All other metals were added into the furnace after reaching the copper melt temperature of 1300 °C. The content of oxygen is controlled in the melt (concentration range was from 1 to 12,000 ppm) and in a copper sample. In the case that the oxygen content was under 200 ppm, the melt was cast into the suitable steel molds. After the self-cooling process, the mechanical processing was used later, as it aimed to prepare anodes for the electrorefining process. The mass of each anode was approximately 7 kg. A detailed procedure of anode preparation was presented previously [17].

#### 2.1.2. Working Electrolyte

The waste sulfate solution from the commercial copper electrolysis AURUBIS AG, Hamburg, Germany (earlier Norddeutsche Affinerie AG), of the following chemical composition (g/L): Cu—32.5; Ni—20.5; As—4; Sb—0.3; Sn—0.001; Pb—0.004, was used as a working electrolyte. Based on the values of concentration of the same elements in the commercial electrorefining process, Ni concentrations were increased and Cu concentration was decreased [18]. The aim was to avoid a sludge precipitation during the electrolyte self-cooling. The samples for analysis were specially prepared, explained in the previous paper [17].

#### 2.1.3. Surfactants

Thiourea and gelatin were used as a surfactant. Thiourea, $CH_4N_2S$, purity grade (purchased from Sigma Aldrich, Company: Merck KGaA, Frankfurter Str. 250, D-64271 DARMSTADT), with a molecular weight of 76.12 g/mol, water solubility of 137 g/L at 20 °C, was used for the minimization of the nodules, porosities, and dendrites' appearance. The solution was prepared with the 18 MΩ cm deionized water.

Gelatin, purity grade, (purchased from Sigma Aldrich, Company: Merck KGaA, Frankfurter Str. 250, D-64271 DARMSTADT) was also used for minimizing the nodules, porosities, and dendrites' appearance to decrease the copper surface roughness. The preparation of the appropriate solution was made with 18 MΩ cm deionized water at a temperature of about 40 °C.

2.1.4. Chemicals for Preliminary Testing the Electrolyte Mixing System

Potassium iodide (purchased from Company: Merck KGaA, Frankfurter Str. 250, D-64271 Darmstadt, Germany) was used for the preparation of 10 wt.% aqueous solution (KI) that was used as electrolyte during preliminary testing of the system for electrolyte mixing.

Sodium thiosulfate ($Na_2S_2O_3$.), purchased from Carl Roth GmbH + Co KG Schoemperlenstr. 3–5, D-76185 Karlsruhe, Germany), was used for the preparation of the aqueous solution for the decolorization of the KI solution.

The preparation of the appropriate solutions was performed with 18 MΩ cm deionized water.

*2.2. Methods*

2.2.1. Experimental Set-Up and Procedure

Electrorefining tests were conducted at the IME Process Metallurgy and Metal Recycling, Aachen, Germany, on the enlarged laboratory equipment, under the constant galvanostatic pulse conditions with the applied current density of 250 A/m² [19–21]. An external source with characteristics of 50 A and 10 V (model HEINZINGER TNB-10-500, Heinzinger Electronic GmbH the Power Supply Company, Rosenheim, Germany) was used as a direct current supplier. The starting cathode sheets were made of stainless steel. Cathode copper (99.95 wt.% Cu) was used as a reference electrode.

A rectangular electrolytic cell made of polypropylene (PP) was used for the electrorefining process. The electrode arrangement in the cell was cathode–anode–cathode, and the electrolyte working volume was 5.85 dm³. The bus bar from copper provided a direct current supply. The cell was current connected with the system for automatic measurement and data processing. A stainless steel water tank with recalculated hot water was used to maintain the required electrolyte temperature in a cell inserted in this tank. The water tank was insulated with styrofoam/aluminum material and covered with a portable cover with openings for the cells. A plastic hose with a manual valve was used for regulation of the water recirculation and circulation speed. The overflow was made through the overflow box at the top of the water tank. Thermostat, HAAKE B7—PHOENIX 2 (Thermo Fisher Scientific, Waltham, MA, USA) was used to maintain the heating water operating temperature. The flow pump was used for water recirculation. Two pumps, BVP Standard (No. ISM 444 ISMATEC, Cole-Parmer GmbH, Wertheim, Germany), were used for dosing an aqueous solution of surfactants, as well as for dosing the deionized water for evaporation in cells. A more detailed description of the experimental setup has been presented previously [17]. The consumption of thiourea and gelatin (50 mg per 1 t of cathode copper, respectively) was in accordance with the literature data, where the best results were obtained when the ratio of those two components was in the range of 0.8 to 1.7 [4]. The aqueous solution of surfactants and deionized water were dosed into a cell continually over 72 h.

The nitrogen distribution system was used for electrolyte mixing. The introduction of nitrogen into a cell was achieved through a glass tube with a diameter of Ø 2 mm, which was placed in the cell, and mixing of electrolytes was achieved by bubbles of solution through a tube with a diameter of Ø 8 mm. Preliminary tests were conducted with two types of nitrogen inlet. In a case (a), a tube with a smaller diameter of 360 mm was placed inside the tube of larger diameter, and in case (b), the tube of smaller diameter was outside and that of the length of 20 mm entered through a lower hole of a tube of larger diameter to the tube itself. In both cases, nitrogen moves from the bottom up through a tube of larger diameter, ensuring the movement of solution in the same direction. The flow of nitrogen is regulated by the existing rotameters. The preliminary tests showed that mixing the electrolyte by introducing nitrogen, as shown in Case (b). gave satisfactory results. The preliminary tests were performed in an existing electrochemical cell with an anode made of PVC material and aluminum cathodes. Synthetic aqueous KI solution was used as an electrolyte. The decolorization course of an aqueous KI solution was tested with the addition of 0.1 mL of $Na_2S_2O_3$. At the same time, nitrogen was introduced through a tube to mix the solution. The color of KI aqueous solution before the addition of $Na_2S_2O_3$

solution was blue. The change in color of the solution from the top to the bottom was observed after the addition of $Na_2S_2O_3$ solution and the introduction of nitrogen in order to mix the solution. The chemical reaction was completed after complete decolorization of the solution. The test results confirmed that the proposed method of introduction of nitrogen into the cell enables the circulation and mixing of the solution.

### 2.2.2. Characterization Methods

The Electro-Nite system (model HERAEUS, Heraeus Electro-Nite International N.V. Centrum Zuid Houthalen, Belgium) was used to control the oxygen content in the copper mixture melt.

The oxygen content in the copper mixture samples was controlled using the Juwe On-mat 8500 instrument (Ströhlein ON-mat 8500, JUWE Laborgeräte GmbH, Germany instrument).

Chemical analysis of copper anodes was done by the Spector Xepos Energy Dispersive X-ray Fluorescence Spectroscope (ED-XRF, SPECTRO, Kleve, Germany). The X-ray anode material was Au. The major and minor elements were determined via the fusion method (1000 °C for 1 h with a mixture of $Li_2B_4O_7/KNO_3$ followed by direct dissolution in 10% $HNO_3$ solution). Samples of copper anodes were obtained by cutting a part of the anode before polishing from the top, middle, and bottom.

The atomic emission spectrometry with inductively-coupled plasma technique (ICP-AES) (model Spectro Ciros Vision, SPECTRO Analytical Instruments GmbH, Kleve, Germany) was used for the determination of the chemical composition of electrolyte samples.

The inductively coupled plasma mass spectrometry (ICP-MS) (model Agilent 7700, Agilent Technologies, Inc., Tokyo 192-8510 Japan) was used for the determination of the element content in anode slime.

Each chemical analysis was carried out in duplicate and accompanied by a quality control (blank and certified reference materials (CRM) analysis).

The X-ray diffraction analysis (XRD) (model Explorer, G.N.R., Via Torino, Italy) was used for the anode slime phase analysis.

### 3. Results and Discussion

*3.1. Anode Chemical Composition*

The results of standard chemical analysis of copper anodes were carried out on 26 elements (Table 1). Samples for the analysis were taken from the bottom, middle, and top of anodes aimed to check the uniformity of anode materials.

**Table 1.** Chemical composition of copper anodes with Ni, Pb, Sn, and Sb non-standard chemical content.

| Element | Position of Anode Sampling | | | | | | | |
|---|---|---|---|---|---|---|---|---|
| | **Bottom** | **Middle** | **Top** | **Average** | **Bottom** | **Middle** | **Top** | **Average** |
| | **Anode with Samples Code An-1** | | | | **Anode with Samples Code An-2** | | | |
| | **Content, wt.%** | | | | | | | |
| Ni | 4.66 | 4.68 | 4.67 | 4.67 | 10.04 | 10.11 | 9.96 | 10.04 |
| Pb | 0.417 | 0.426 | 0.419 | 0.421 | 0.392 | 0.363 | 0.400 | 0.385 |
| Sn | 0.443 | 0.445 | 0.444 | 0.444 | 0.412 | 0.395 | 0.419 | 0.41 |
| Sb | 0.443 | 0.450 | 0.449 | 0.447 | 0.382 | 0.369 | 0.394 | 0.382 |
| Zn | <0.0015 | <0.0015 | <0.0015 | <0.0015 | 0.0055 | 0.0058 | 0.0060 | 0.0058 |
| P | 0.0052 | 0.0052 | 0.0051 | 0.0052 | 0.0065 | 0.0064 | 0.0064 | 0.0065 |
| Fe | 0.016 | 0.016 | 0.014 | 0.0153 | 0.014 | 0.016 | 0.015 | 0.015 |
| Si | 0.0047 | 0.0056 | 0.0024 | 0.0042 | 0.010 | 0.0031 | 0.0027 | 0.0012 |
| Cr | <0.0003 | <0.0003 | <0.0003 | <0.0003 | 0.0005 | 0.0006 | 0.0006 | 0.0006 |
| Te | 0.0064 | 0.0075 | 0.0075 | 0.007 | 0.0020 | 0.0024 | 0.0022 | 0.0022 |
| As | 0.028 | 0.029 | 0.029 | 0.0287 | 0.021 | 0.020 | 0.020 | 0.020 |
| Cd | 0.0019 | 0.0019 | 0.0018 | 0.0019 | 0.0010 | 0.0011 | 0.0011 | 0.0011 |
| Bi | 0.0038 | 0.0036 | 0.0035 | 0.0036 | 0.0027 | 0.0028 | 0.0028 | 0.0028 |
| Ag | 0.068 | 0.066 | 0.067 | 0.067 | 0.053 | 0.055 | 0.053 | 0.054 |

**Table 1.** *Cont.*

| Element | Position of Anode Sampling | | | | | | | |
|---|---|---|---|---|---|---|---|---|
| | Bottom | Middle | Top | Average | Bottom | Middle | Top | Average |
| | Anode with Samples Code An-1 | | | | Anode with Samples Code An-2 | | | |
| | Content, wt.% | | | | | | | |
| S | 0.0031 | 0.0032 | 0.0031 | 0.0031 | 0.0048 | 0.0048 | 0.0049 | 0.0049 |
| Au | <0.0005 | <0.0005 | <0.0005 | <0.0005 | 0.0027 | 0.0018 | 0.0018 | 0.0018 |
| C | 0.020 | 0.011 | 0.020 | 0.0300 | 0.025 | 0.032 | 0.031 | 0.029 |
| Ti | 0.0017 | 0.0017 | 0.0017 | 0.0017 | 0.0023 | 0.0027 | 0.0025 | 0.0025 |
| Se | 0.0060 | 0.0061 | 0.0060 | 0.0060 | 0.0078 | 0.0079 | 0.0079 | 0.0079 |

Close values for the content of elements from different sampling positions confirmed the efficiency of the anode preparation process. Copper content was calculated as a difference of Ni, Pb, Sn, and Sb average values up to 100 wt.%. The calculated value of Cu content in the anodes with sample codes An-1 and An-2 was 94.02 wt.% and 88.78 wt.%, respectively. Additionally, Table 1 presents the average values for all elements obtained on the basis of mathematical calculation for three values for each element. The results of chemical analyses have shown that the content of Zn, Mn, Mg, Cr, Co, Al, Be, Zr, Au, and B in the anode with sample code An-1 was below the sensitivity limit of the used analytical method. For the anode with sample code An-2, the content of Zn, Cr, and Au had the values over the sensitivity limit of the used analytical method, but the content of Mn, Mg, Co, Al, Be, Zr, and B was below the sensitivity limit of the used analytical method for the anode with sample code An-1. The content of P, Fe, Si, Cd, Bi, Ag, S, C, Ti, and Se in both anode types was within the limits in industry practice [16]. The obtained results for oxygen content in anode An-1 and An-2 had values of 120 and 98 ppm, respectively. Those values are in accordance with the investigation realized by the authors [22] that confirmed the aim to decrease the content of oxide and "kupferglimmer" forms in anodes with oxygen content to be lower than 200 ppm.

### 3.2. Anode Dissolution and Cathode Deposit

The mass of anodes was measured at the beginning and end of the process. Dissolved anode mass was calculated as the difference between those two values. Table 2 presents the values for the mass of dissolved elements from anodes during 72 h of the experiments at different electrolyte temperatures.

**Table 2.** Mass of dissolved copper anodes and cathode deposit mass.

| Anode with Samples Code An-1 | | Anode with Samples Code An-2 | |
|---|---|---|---|
| Electrolyte working temperature | | | |
| T1 = 63 $\pm$ 2 °C | T2 = 73 $\pm$ 2 °C | T1 = 63 $\pm$ 2 °C | T2 = 73 $\pm$ 2 °C |
| Dissolved anode mass, g | | | |
| 1688 | 1779 | 1367 | 1442 |
| Cathode correspond to anode An-1 | | Cathode correspond to anode An-2 | |
| Cathode deposit mass, g | | | |
| 1708 | 1789 | 1371 | 1526 |

Cathode mass as a cumulative mass of starting cathode sheet and deposited material after 72 h was measured at the end of the process. The difference between the mass of the starting cathode sheet and the cumulative cathode mass at the end of the process was the mass of material deposited during the process (Table 2).

Data from Table 2 show that the mass of the cathode deposit is higher than the dissolved mass of corresponded anode. It could be explained by the electrowinning

process of Cu ions from the working electrolyte (Table 3). By the electrowinning process, copper is deposited directly from the working electrolyte onto a cathode sheet under the action of the direct current. General electrochemical reactions for this process are shown by the individual reactions on the cathode and anode as well as by the cumulative reaction in the system [23]:

**Table 3.** Electrolyte chemical composition changing.

| | Anode with Samples Code An-1 | | | | | |
|---|---|---|---|---|---|---|
| | T1 = 63 ± 2 °C | | | T2 = 73 ± 2 °C | | |
| **Element** | Process Duration, h | | | | | |
| | 24 | 48 | 72 | 24 | 48 | 72 |
| | Element Concentration, g/L | | | | | |
| Cu | 23.8 | 18.5 | 10.9 | 22.5 | 17.6 | 2.2 |
| Ni | 25.6 | 29.4 | 32.7 | 25.5 | 31 | 36.5 |
| As | 3.7 | 2.7 | 1.9 | 3.5 | 2.6 | 1 |
| Sb | 0.292 | 0.290 | 0.217 | 0.267 | 0.237 | 0.156 |
| | **Anode with Samples Code An-2** | | | | | |
| | T1 = 63 ± 2 °C | | | T2 = 73 ± 2 °C | | |
| **Element** | Process Duration, h | | | | | |
| | 24 | 48 | 72 | 24 | 48 | 72 |
| | Element Concentration, g/L | | | | | |
| Cu | 21 | 10.4 | 6.8 | 18 | 12.5 | 1.5 |
| Ni | 28.5 | 33.5 | 43.6 | 31 | 41 | 47 |
| As | 3.6 | 2.8 | 0.85 | 3.5 | 2.5 | 0.8 |
| Sb | 0.295 | 0.292 | 0.24 | 0.294 | 0.288 | 0.188 |

Cathode reaction:

$$Cu^{2+} + 2e \rightarrow Cu^{0}, \ E^{0} = +0.34 \ V \tag{1}$$

Anode reaction:

$$H_2O \rightarrow H^{+} + (OH)^{-} \rightarrow \frac{1}{2}O_2 + 2H^{+} + 2e, \ E^{0} = +1.23 \ V \tag{2}$$

Cumulative reaction:

$$Cu^{2+} + SO_4^{2-} + H_2O \rightarrow Cu^{0} + \frac{1}{2}O_2 + 2H^{+} + SO_4^{2-}, \ E^{0} = -0.89 \ V \tag{3}$$

The dissolved anode mass had the lowest value for the anode with the sample code An-2 after electrorefining at the working electrolyte temperature of T1 = 63 ± 2 °C (Table 2). Masses of dissolved anodes had higher values for anodes with lower content of ingredients. In the case when the value for the average total content of Ni, Pb, Sn, and Sb was 5.982 wt.% (anode An-1), the mass of the dissolved anode was higher for 321 g in comparison with the dissolved mass of anode An-2 (average value for the total content of ingredients was 11.217 wt.%) for the same process condition. When the Ni content in anodes is lower than 3 wt.%, Ni dissolution in the electrolyte is near 100%. In a case that the Ni content is higher than 3 wt.%, the ingredients are present in the working electrolyte as the Ni and Cu-Ni-Sb oxide forms and $NiSO_4$. Ni salt ($NiSO_4$) participates in a transmission of electricity and creates the unfavorable conditions for the discharge of copper ions, reducing its concentration in the vicinity of the cathode. In addition, increasing the concentration of ballast salts in the electrolyte reduces its electrical conductivity and the dissolution of elements.

Cell Voltage Changing

During the electrorefining process, the cell voltage was measured every 10 s. The cell voltage changing for different anode types and different working electrolyte temperatures is defined on the basis of about 25,000 data (Figures 1 and 2).

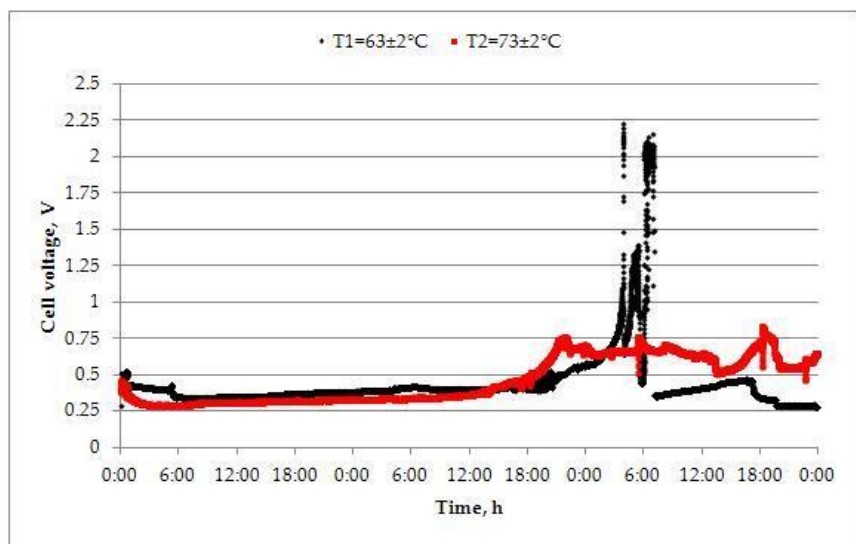

**Figure 1.** Cell voltage changing during the electrorefining of anode with sample code An-1, for different working electrolyte temperatures, 72 h duration.

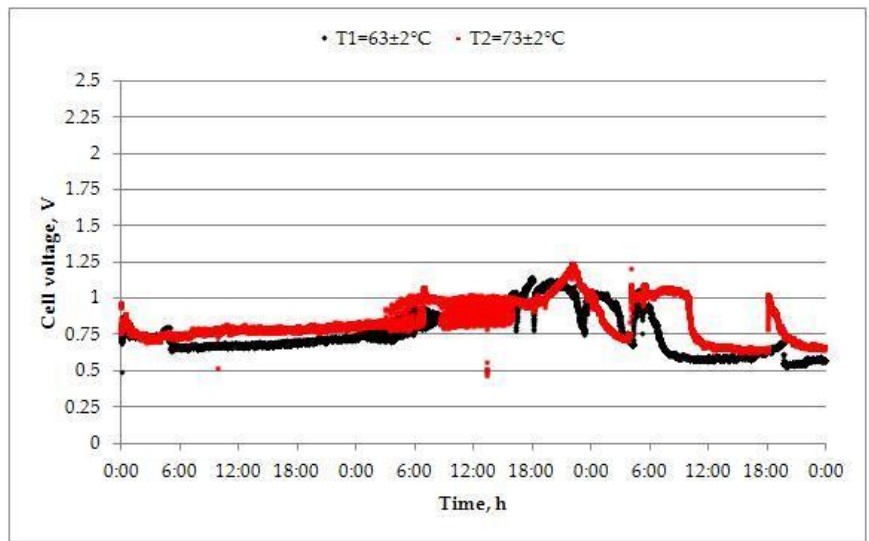

**Figure 2.** Cell voltage changing during the electrorefining of the anode with sample code An-2, for different working electrolyte temperatures, 72 h duration.

The changing of the cell voltage during each test (Figures 1 and 2) was discussed on the basis of several characteristic changes of cell voltage during the non-commercial electrorefining process [24]. Further explanation of data from Figures 1 and 2 will be appointed for the first appearance points of each phase.

From Figure 1 which that corresponds to the anode with sample code An-1, the time of active dissolution was 42:19 h (T1 = 63 ± 2 °C). During that time period, the cell voltage oscillation was in the range of 0.2 V. After the end of the first phase period, which is called the "active dissolution phase", the value for cell voltage oscillation was more than 0.2 V, and this phase was called the "oscillation phase". The duration of the first appearance of the oscillation phase was approximately 2 h for the test on T1 = 63 ± 2 °C. The phase when

the cell voltage suddenly was increased is called the "occurrence of full passivation". The first appearance of the suddenly increased cell voltage (2.22 V, Figure 1) is registered after 51:49 h (anode An-1) on T1 = 63 ± 2 °C. The cell voltage changing at a higher working electrolyte temperature (T2 = 73 ± 2 °C) had asimilar trend but without the appearance of the passivation peak. It is confirmed by a higher value of dissolved anode mass (Table 2).

It is characteristic that the total anode passivation time for the anode with sample code An-1 was a few minutes, and the lower value of the dissolved anode mass at the working electrolyte temperature (T1 = 63 ± 2 °C) can be explained by a layer of anode slime on the anode surface. Therefore, for the test at higher temperature (T2 = 73 ± 2 °C), the cell voltage oscillation of more than 0.2 V was not registered, but an increase of cell voltage has appeared at the same time when the passivation peak at a lower working electrolyte temperature (T1 = 63 ± 2 °C) appeared. The electrorefining process continues on the increased value of cell voltage (0.75 V). This value of cell voltage could be explained as a consequence of particular anode passivation (Figure 1). The "full-time passivation phase" could be explained as a phase with a constant high cell voltage value, and this phase does not appear in Figure 1.

The results from Figure 2 have indicated almost identical behavior of the anode with sample code An-2 during the electrorefining process at different working electrolyte temperatures. Electrorefining processes are carried out on the cell voltage from approximately 0.6 to 1.26 V (Figure 2). As a consequence of particular anode passivation, the electrowinning of Cu ions from the electrolyte at both working electrolyte temperatures occurred (Table 3). Additionally, the dissolved anode mass was lower than for the anode with sample code An-1. The cell voltage for the stable phase also had similar values as the anode with sample code An-1. For the test realized at T1 = 63 ± 2 °C, this value was stable at about 25 h, and for the test at T2 = 73 ± 2 °C, the duration of the stable phase was about 26 h (Figure 2). Additionally, the mass of anode slime and Cu content in anode slime confirmed no evident difference during the electrorefining process of the anode with sample code An-2 at a different working electrolyte temperature (Table 4).

**Table 4.** Anode slime chemical composition, mass of anode slime, and anode slime mass percentage in relation to the dissolved anode mass.

| Anode Type | Anode with Sample Code An-1 | | Anode with Sample Code An-2 | |
|---|---|---|---|---|
| Working Electrolyte Temperature | T1 = 63 ± 2 °C | T2 = 73 ± 2 °C | T1 = 63 ± 2 °C | T2 = 73 ± 2 °C |
| Element | Element content, wt.% | | | |
| Cu | 13.1 | 14.7 | 59.0 | 59.8 |
| Ni | 3 | 3.2 | 1.2 | 1.4 |
| Pb | 29.2 | 31.9 | 17.5 | 16.9 |
| Sn | 2.8 | 3.1 | 1.3 | 1.4 |
| Sb | 19.7 | 17.6 | 6.3 | 6.4 |
| As | 6.2 | 5.8 | 2.5 | 2.1 |
| **Total Mass of Anode Slime, g** | | | | |
| Total Mass of Slime from the Anode Surface and Cell Bottom | 59.9 | 63.06 | 88.57 | 72.24 |
| Anode Slime Mass Percentage in Relation to the Dissolved Anode Mass, wt.% | | | | |
| Total Anode Slime Mass Percentage | 3.55 | 3.58 | 6.47 | 5.01 |

In relation to the kinetics, the cell voltage and current are theoretically connected over the Tafel approximation of the electrochemical kinetic equation (Butler–Volmer). Once the exchange current density and Tafel slope is known for copper dissolution/deposition, the overpotentials on the cathodic and anodic side can be readily calculated at the known temperature of the refining process. Being of a large exchange current density and low Tafel slope, the copper electrochemical reaction is of rather small overpotentials of several hundreds of mV. However, the hydrodynamics also affects the overpotential, which is hard

to calculate for a given cell geometry and can be only gained empirically by the independent measurements, e.g., current vs. flow. In addition, the cell voltage also comprises the ohmic drop within the electrolyte, which can change as the process goes on, and the voltage drops at electric contacts and within the electrodes, which are to be arbitrarily set to reach the measured cell voltage. It follows that any measured cell voltage can be easily compared to the theoretically calculated by assigning the hardly measurable drops (electrolyte, contacts, electrodes, etc.) to the difference between measured and calculated values, which has nothing to do with the refining process itself or to the type of dissolving anode as the scopes of this manuscript. Comparison is important if some specific cell geometry and construction have to be chosen, but in this case, these parameters were fixed in order to investigate the anode dissolution.

### 3.3. Working Electrolyte Chemical Composition Changing

A real sulfuric waste solution of the following chemical composition (g/L): Cu—32.5; Ni—20.5; As—4; Sb—0.3; Sn—0.001; Pb—0.004 was used as the working electrolyte. Each sample analyzed at 24, 48, and 72 h was taken from the middle of the cells. Sludge precipitation has occurred in samples that were analyzed without any previous preparation. Further, the samples were prepared by the next procedure: 10 mL of the electrolyte from the cell medium, plus 10 mL of concentrated HCl and deionized water up to 50 mL. The appropriated dissolution was used for a higher concentration of the analyzed elements. Monitoring the change in the concentration of Cu, Ni, As, and Sb is presented in Table 3.

Data from Table 3 were used for calculating the changes in Cu, Ni, As, and Sb concentration. The concentrations of those elements in the starting electrolyte were the basic values (100%).

The results presented in Figure 3 have confirmed that the concentration of Cu, As, and Sb ions was decreased during the tests at different temperatures. On the other hand, Ni concentration is increased during the tests. Cu ions' concentration decrease had similar values for both anodes in the same operational conditions (Figure 3a,b). The highest value of Cu ions decreasing is registered for an anode with the sample code An-2 (with Cu content in the anode of 88.78 wt.%). At the same time, the increase in Ni ions' concentration has the highest value of 129.27 wt.% at the end of electrorefining for the same anode. The decrease in Cu ions' concentration in the working electrolyte directly indicates the fact that the electrowinning process of copper ions has occurred at the same time. The identical phenomenon is characteristic for the electrorefining test with anode with the sample code An-1, where Cu content is 94.02 wt.%. At the increased working electrolyte temperature, the values for decreasing Cu, As, and Sb ions concentration were higher in comparison with the values at lower temperature for both anode types.

Literature data have confirmed the negative impact of Ni on the electrorefining process [13,25,26]. Nickel ions' concentration in the working electrolyte up to 25 g/L is a concentration that has no harmful effect on cathode precipitate. However, a high Ni ion concentration in the electrolyte reduces the copper solubility at the anode/electrolyte interface and leads to partial or full anode passivation and interruption of the electrorefining process (Figure 2) [17,24].

The highest decrease in Cu ions' concentration (95.38 wt.%) has been achieved by the electrorefining of the anode with sample code An-2 at the working electrolyte temperature of T2 = 73 ± 2 °C (Figure 3). By comparison of the values for decreasing the Cu ions; concentration at the end of the process for different anode types, it could be seen that the Cu ions' concentration decrease is higher by about 30% during the electrorefining process of the Cu anode with additional 10 wt.% Ni. It is the same for both working electrolyte temperatures.

The same behavior is characteristic for changing the As ions' concentration. The highest decrease in As ions' concentration (80 wt.%) was also registered during the electrorefining process of anode with the sample code An-2 at the working electrolyte temperature of T2 = 73 ± 2 °C (Figure 3).

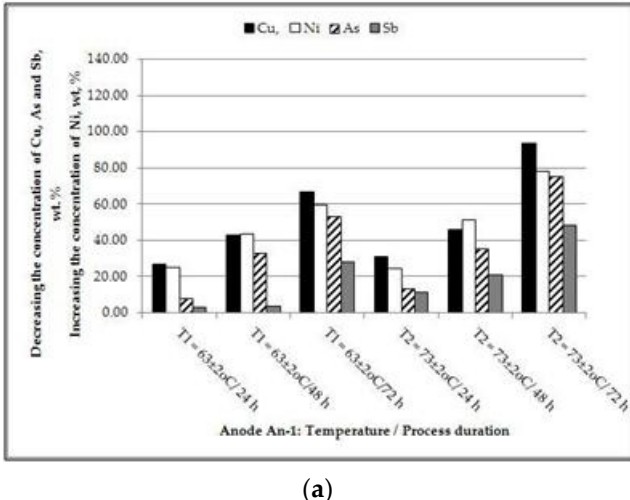

(**a**)

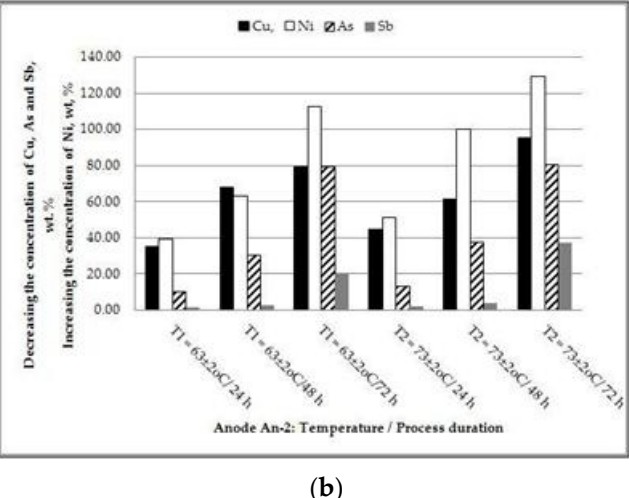

(**b**)

**Figure 3.** Cu, Ni, As, and Sb concentration changing during the electrorefining process for anodes: (**a**) with sample code An-1 and (**b**) with sample code An-2.

The decrease in Sb ions' concentration is also registered for a higher temperature of the working electrolyte (T2 = 73 ± 2 °C), and this value had the maximum value for the anode with sample code An-1 of 48 wt.%.

The increased concentration of Ni ions, as a consequence of Ni electrolytic dissolution from anodes, had the highest value for the anode with sample code An-2 (129.27%). This value is registered at the end of the electrorefining process that was carried out at a working electrolyte temperature of T2 = 73 ± 2 °C. This value is about 30% higher than the value of Ni ions, increasing during the process that was carried out at a lower working electrolyte temperature. In the case of changing the Cu ions concentration, it could be a consequence of the increased dissolution of copper and nickel salts at an elevated temperature. However, a high concentration of nickel in the electrolyte reduces the solubility of copper at the anode–electrolyte interface and leads to the passivation of the anode and interruption of the electrolysis process [27].

*3.4. Anode Slime*

The slime formed during the electrorefining process of Cu anodes with non-commercial chemical composition is a result of electrochemical and chemical processes onto the anodes and in the working electrolytes. The slime originates from the anode surface and from the bottom of the electrolytic cell. At the end of each test, the anode slime was stripped from the anode surfaces, separately filtered, and after washing and drying, individually measured in order to compare the mass of slime present in the working electrolyte. The sample used for chemical characterization was a composite sample from the anode surface and electrolytic cell. The anode slime sample of 0.5 g was dissolved with aqua regia, transferred into a volumetric flask of 50 mL, and completed with deionized water. The appropriated dissolution was used for a higher concentration of analyzed elements.

The results from Table 4 show that the anode slime makes up 3.55–6.47 wt.% of dissolved anodes mass. Those values are higher than in the commercial electrorefining copper anodes process, where the anode slime makes up 0.2–0.8 wt.% [28].

The Cu content has a higher value in the anode slime obtained during the electrorefining of the anode with sample code An-2. Decreasing the Cu ions' concentration in the working electrolyte has a negative impact on the morphology of the cathode deposit. Additionally, the elementary copper fell off from a cathode sheet as a consequence of the non-compact cathode deposit (Figure 2 and data from Table 3) [17]. Values for the cell voltage during the electrorefining tests had a similar value in both cases (Figure 2). The oscillation phase had similar values, and cell voltage had a higher value than the values in the standard copper electrorefining process [29]. The maximum reduction of Cu ions' con-

centration in the working electrolyte is registered for the anode with the sample code An-2 during the electrorefining process at higher electrolyte temperature (Table 3). The results from Table 3 confirmed that the electrowinning of Cu ions from the working electrolyte is also carried out during 72 h of tests. The cathode deposit has different characteristics in comparison with the standard Cu electrorefining process, and a part of cathode copper was dropped into anode slime in a form of copper (I) oxide or as copper powder. The copper content in the anode with the sample code An-1 has a lower value, and it is in accordance with changing the Cu ions concentration in the working electrolyte during each test. The presence of nickel in anode slime is explained as a consequence of electrolyte inclusion in anode slime. Sn content has similar values for both anode types. The Sb/As ratio is about 3/1 for each anode slime sample.

XRD Analysis of Anode Slime

The X-ray diffraction (XRD) (model Explorer, G.N.R. srl, Via Torino, Italy) was used for the anode slime phase analysis. The device uses CuK$\alpha$ X-ray radiation of wavelength 1.54 Å. The X-ray tube voltage is 40 kV, and current consumption is 30 mA. The detector is a scintillation counter, and the geometry of the device is $\theta$-$\theta$. Each analysis lasted 40 min. Match!, Version 2 and "Crystallography Open Database" COD REV44788 databases were used for the phase identification from the powder diffraction. The Match! identifies a phase in a sample by comparing its powder diffraction pattern to the reference patterns of known phases. Hence, it needs a so-called "reference database" in which these reference patterns are provided. The Match! is extremely flexible in this context, with several options for obtaining/using a reference database. A reference database based on the COD is installed automatically along with Match!. This reference database contains the powder diffraction patterns calculated from the crystal structure data taken from the COD, which itself provides the crystal structure data published by the IUCr journals, the "American Mineralogist Cristal Structure Databases" (AMCSD), and various other sources. All entries taken from the COD reference database contain the atomic coordinates, based on which the corresponding powder diffraction patterns have been calculated. Besides this, the I/Ic-values have been calculated for all entries, so that a semi-quantitative analysis can be carried out. Each sample is recorded in the interval from 10° to 70° 2$\theta$ with a step of 0.05° 2$\theta$. The anode slime samples were prepared in a powder form of a particle size of 5 to 10 microns in an agate mortar.

The quantitative results are the results of a software evaluation that is automatically calculated as each new phase is entered into the analysis results. The Match! software package, version 2.0 with the COD reference database was used for identifying the present phases and determining their content in the anode slime obtained during the electrorefining process of the anode with the sample code An-1 and An-2. The phase identification was performed by comparing the obtained data to data from a standard database. The content of the phases was calculated using the software and as such can be used as the relative data to compare the number of phases.

The results of each tested anode slime sample (from anode with the sample code An-1 and An-2), the corresponding patterns (Figures 4–7), and identified phases with formulas and names, expressed in the weight percent (wt.%) (Table 5), are presented. Peaks that belong to one phase are denoted by the same letter symbol.

It can be seen from Figure 4a that the major crystalline phases in the homogenized sample of anode slime were Anglesite ($PbSO_4$), Copper(I) arsenide Domeykite low ($Cu_3As$), Antimony arsenate ($SbAsO_4$), Copper(I) oxide Cuprite ($Cu_2O$), Claudetite ($As_2O_3$), Litharge (PbO), and Senarmontite ($Sb_2O_3$).

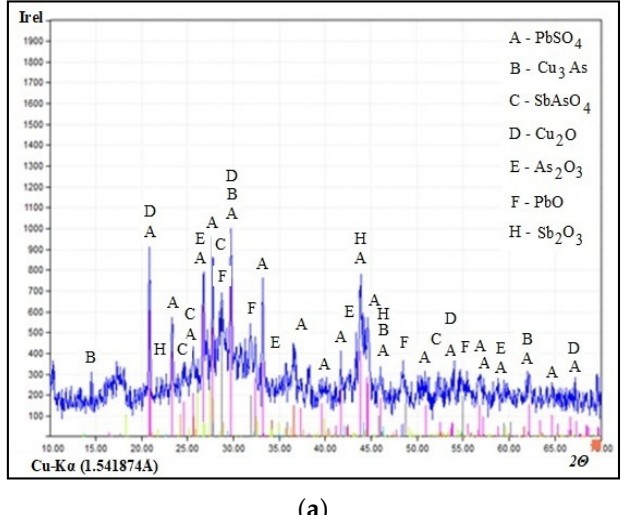

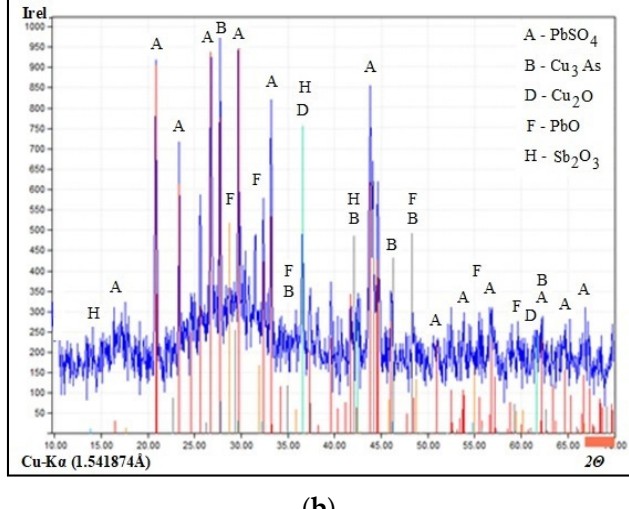

**Figure 4.** X-ray diffraction patterns of homogenized slime after electrorefining of anode with the sample code An-1: (**a**) working electrolyte temperature of T1 = 63 ± 2 °C and (**b**) working electrolyte temperature of T2 = 73 ± 2 °C.

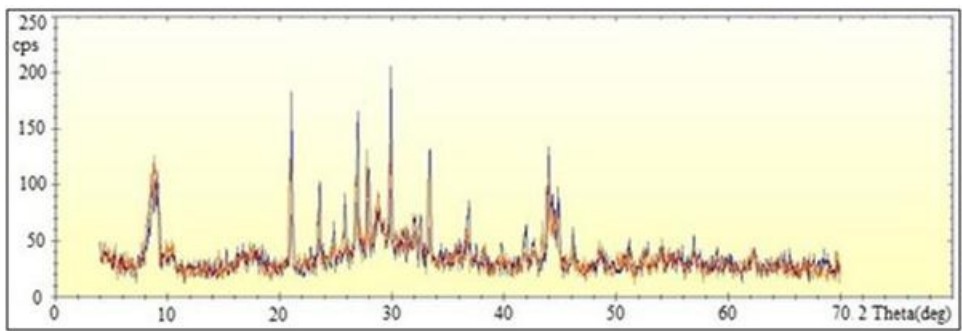

**Figure 5.** Comparative presentation of anode slime phase analyzes. Red represents the patterns at a temperature of T1 = 63 ± 2 °C, and blue represents the patterns at a temperature of T2 = 73 ± 2 °C for the anode with sample code An-1.

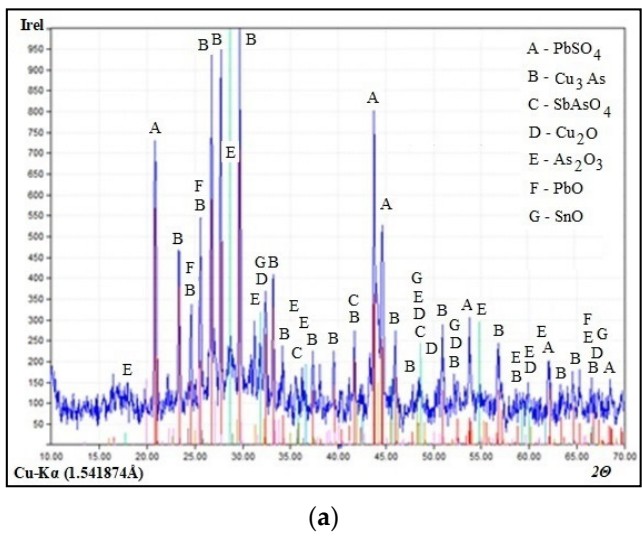

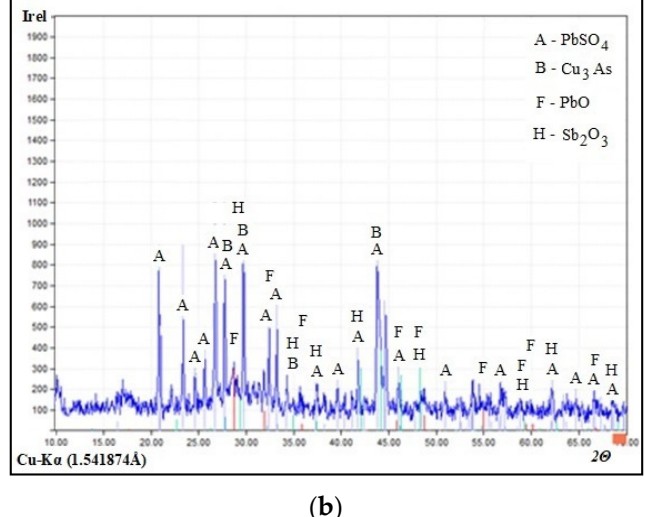

**Figure 6.** X-ray diffraction patterns of homogenized slime after the electrorefining of the anode with sample code An-2: (**a**) working electrolyte temperature of T1 = 63 ± 2 °C and (**b**) working electrolyte temperature of T2 = 73 ± 2 °C.

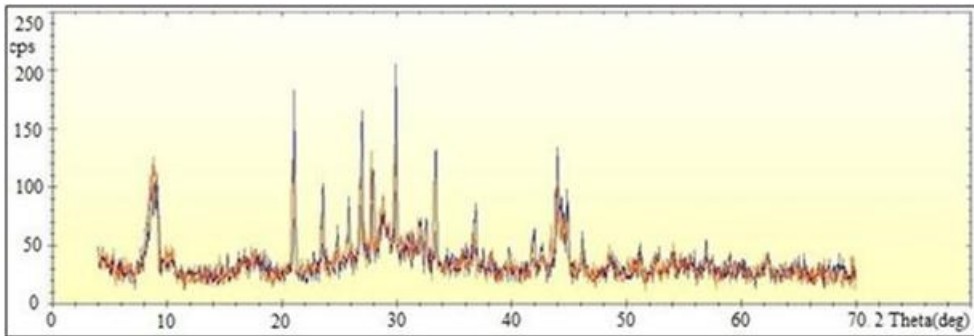

**Figure 7.** Comparative presentation of anode slime phase analyses. Red represents the patterns at a temperature of T1 = 63 ± 2 °C, and blue represents the patterns at a temperature of T2 = 73 ± 2 °C for the anode with sample code An-2.

With the increase in the working electrolyte temperature, the major crystalline phases were reduced on $PbSO_4$, $Cu_3As$, $Cu_2O$, PbO, and $Sb_2O_3$. PbO peaks intensity is decreased in the anode slime obtained by the electrorefining process at higher working electrolyte temperature. From data presented in Table 5, the phase $As_2O_3$ is present at a value of 30.4 wt.% and $SbAsO_4$ in value of 7.5 wt.% for anode slime obtained at a temperature of T1 = 63 ± 2 °C. Those phases are not registered in slime obtained from the same anode at temperature of T2 = 73 ± 2 °C. The chemical composition of anode slime confirms that the amount of As and Sb at a higher temperature is lower compared to the values at lower working electrolyte temperature (Table 4).

Based on the patterns from Figure 4, $PbSO_4$ has a more crystalline form in anode slime obtained at a lower temperature. Clusters at lower temperatures tend to reach the crystalline form for copper. Cu (I) in the form of $Cu_2O$ and $Cu_3As$ is more present in anode slime obtained at higher working electrolyte temperature based on the fact that the copper clusters achieved a more difficult crystalline form at lower temperatures (Table 5) and based on the fact that the concentration of Cu(I) ions in the electrolyte increases with the increase in the working electrolyte temperature [30]. Because of that, the more complete crystallization of Cu (I) ions was achieved at a temperature of T2 = 73 ± 2 °C. This is also confirmed by the values of the estimated mass percentage of these compounds given in Table 5.

**Table 5.** Estimated percentage composition of the registered phases in anode slime from different types of electrorefining processes.

| Symbol | Pattern Numbers | Phase Name | Chemical Formula | Anode with Samples Code An-1, (wt.%) | | Anode with Samples Code An-2, (wt.%) | |
|---|---|---|---|---|---|---|---|
| | | | | T1 = 63 ± 2 °C | T2 = 73 ± 2 °C | T1 = 63 ± 2 °C | T2 = 73 ± 2 °C |
| A | [96-900-0653] [96-900-5525] | Anglesite | $PbSO_4$ | 45.0 | 48.4 | 55.0 | 75.9 |
| B | [96-101-0976] | Copper(I) arsenide Domeykite low | $Cu_3As$ | 4.9 | 31.2 | 20.2 | 20.7 |
| C | [96-901-1215] | * Antimony Arsenate | $SbAsO_4$ | 7.5 | - | 16.1 | - |
| D | [96-101-0927] [96-101-0964] | Copper(I) oxide Cuprite | $Cu_2O$ | 4.2 | 15.4 | 6.5 | - |
| E | [96-900-9692] [96-900-9694] | Claudetite | $As_2O_3$ | 30.4 | - | 1.2 | - |
| F | [96-901-2701] [96-901-2700] | Litharge | PbO | 6.5 | 3.9 | 0.9 | 2.4 |
| G | [96-110-1036] | Cassiterite | SnO | - | - | 0.2 | - |
| H | [96-900-9748] | Senarmontite | $Sb_2O_3$ | 1.2 | 1.2 | - | 1.0 |

* The existence of $SbAsO_4$ form was emphasized in the papers [15,31] as presented in those papers.

The presence of Ni phases was not observed. This indicates that the Ni content in anode slime is very low, which is confirmed by the results of anode slime chemical composition given in Table 4.

Figure 5 presents a comparative presentation of the phase analyses of anode slime obtained during the electrorefining processes of the anode with sample code An-1 at different working electrolyte temperatures.

Overlapping patterns confirmed that the phases are better crystallized at a higher temperature. The appearance of higher concentrations of $Cu_3As$ and $Cu_2O$ gave the higher peak intensities at a higher temperature of T2 = 73 $\pm$ 2 °C, and this was confirmed with the results given in Table 5.

Figure 6 shows a phase diagram of anode slime obtained by the electrorefining processes of the anode with sample code An-2 at temperatures (a) T1 = 63 $\pm$ 2 °C and (b) T2 = 73 $\pm$ 2 °C. Characteristic for those patterns is a large number of peaks and overlapping the peaks of some phases. $PbSO_4$, $As_2O_3$, $SbAsO_5$, $Cu_3As$, $Cu_2O$, PbO, SnO, and $Sb_2O_3$ are detected in anode slime and presented by the X-ray diffraction patterns in Figure 6a. The next phases, $PbSO_4$, $Cu_3As$, PbO, and $Sb_2O_3$, are detected in the anode slime sample obtained at T2 = 73 $\pm$ 2 °C (Figure 6b). $SbAsO_4$, $Cu_2O$, and $As_2O_3$ are presented in the mass percentage of 16.1, 6.5, and 1.2, respectively, only in the anode slime obtained at a temperature of T1 = 63 $\pm$ 2 °C (Table 5, Figure 6b).

The main characteristic for patterns in Figures 4 and 6 is that $SbAsO_4$ and $As_2O_3$ are not registered in the anode slime samples obtained during the electrorefining of anodes with sample code An-1 and An-2 at higher working electrolyte temperature (T2 = 73 $\pm$ 2 °C).

Figure 7 gives a comparative presentation of the phase analyzes of anode slime obtained during the electrorefining processes of the anode with sample code An-2 at different working electrolyte temperatures.

Overlapping patterns indicated that the phases are better crystallized at a higher temperature. The appearance of higher concentrations of $PbSO_4$, $Cu_3As$, and PbO gave the higher peak intensities at a higher temperature, and it is confirmed by the results given in Table 5. The results shown in Table 5 are semi-quantitative, and those data were not the exact data. Those data were used in order to compare the characteristics of anode slime obtained from the same non-standard copper anode at different temperatures.

The amorphous part of the material obtained during the electrorefining processes of anodes could not be discussed, although a form of X-ray patterns confirmed the presence of amorphous material, so that the percentage composition of crystallized phase is not a precise figure, but estimates by the software automatically calculate and apply only for the crystalline phase. The formation of the floating slimes with the amorphous characteristics was associated with oxidation of Sb(III) to Sb(V) [32,33]. For future investigation, it is possible to estimate the amorphous phase content, for example, by adding a known amount of an internal standard sample (e.g., silicon). The amount of amorphous phase can be estimated from the difference between the exact amount of silicon added and the amount of silicon determined by the XRD.

## 4. Conclusions

Copper anodes of non-commercial chemical composition regarding the Ni, Pb, Sn, and Sb content were prepared to test the electrorefining process. Ni content was 5 or 10 wt.%, and the content of the main ingredients (Pb, Sn, and Sb) was 0.5 wt.% for each one. The real waste solution of sulfuric acid character that was used as a working electrolyte had the Cu ions of decreased concentration in relation to the commercial value. It was confirmed that the electrorefining process of those anodes can be carried out for a limited number of hours. The decrease in the Cu ions' concentration (max. 95.38 wt.%) in the working electrolyte is a limited parameter for electrorefining the non-commercial copper anodes. The concentration of Ni ions increased, and the highest value of 129.27 wt.% was for the anode with a higher Ni content in the anode. The concentration of As ions in the working electrolyte decreased by up to 80 wt.%, and Sb ions' concentration decreased by a maximum of 48 wt.%. It

was also observed that the full passivation did not occur. The masses of cathode deposits confirmed that the Cu ions from the electrolyte were deposited onto cathodes. The mass of anode slime makes up 3.55–6.47 wt.% of the dissolved anode mass. The major crystalline phases in anode slime, detected by the X-ray diffraction analyses, were $PbSO_4$, $Cu_3As$, $SbAsO_4$, $Cu_2O$, $As_2O_3$, PbO, SnO, and $Sb_2O_3$. A larger number of crystalline phases was registered at a temperature of T1 = 63 ± 2 °C.

**Author Contributions:** Individual contributions of authors are as follows: conceptualization, R.M., J.S. and S.S.; methodology, R.M. and B.F.; XRD analysis and software, V.K.; validation, V.M. and Z.S.; formal analysis, V.M. and R.M.; investigation, R.M.; resources, B.F., S.S. and R.M.; data curation, R.M. and V.M.; writing—original draft preparation, R.M., V.K. and V.M.; writing—review and editing, R.M., S.S. and V.M.; visualization, Z.S.; supervision, J.S.; project administration, R.M. All authors have read and agreed to the published version of the manuscript.

**Funding:** Financial support for this study was partly provided by the Ministry of Education, Science and Technological Development of the Republic of Serbia (Contract No. 451-03-9/2021-14/200052).

**Institutional Review Board Statement:** Not applicable.

**Informed Consent Statement:** Not applicable.

**Data Availability Statement:** The data presented in this study are available on request from the corresponding author.

**Conflicts of Interest:** The authors declare no conflict of interest.

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
