# Peer review of "Electrorefining Process of the Non-Commercial Copper Anodes"

_metals, doi:10.3390/met11081187_

Round 1

Reviewer 1 Report

This paper evaluates the electrochemical behavior of copper with impurities higher than those handled in commercial process, and is considered to have great academic and industrial significance in the current industry, where the importance of recycling technology is growing. The manuscript was written logically, but if some data are presented in an electrochemical standard method, more meaningful consideration is possible.

In the case of figure 1,2 of Chapter 3.2.1, cell voltage is shown according to refining time. Even if a quasi Cu reference electrode is used, the source of resistance can be explained more clearly if the anode and cathode potentials are shown separately. 

Also, the authors explain anode passivation with cell voltage data, and although the factor for increasing resistance is on the anode side, it is difficult to determine it with cell voltage.

If there is data that measures the anode and cathode potentials separately, please show them together with the cell voltage.

A more quantitative evaluation is possible if the theoretical dissolution and electrodeposition amounts are displayed together with the weight change of the anode and cathode in Table 2. In addition, current efficiency can be compared, so it will be more meaningful data.

Author Response

Dear Reviewer,

thank you very much for valuable comments and invested time.

We are sending our answers on your comments (bold letters)

This paper evaluates the electrochemical behavior of copper with impurities higher than those handled in commercial process, and is considered to have great academic and industrial significance in the current industry, where the importance of recycling technology is growing. The manuscript was written logically, but if some data are presented in an electrochemical standard method, more meaningful consideration is possible.

In the case of figure 1,2 of Chapter 3.2.1, cell voltage is shown according to refining time. Even if a quasi Cu reference electrode is used, the source of resistance can be explained more clearly if the anode and cathode potentials are shown separately. 

Also, the authors explain anode passivation with cell voltage data, and although the factor for increasing resistance is on the anode side, it is difficult to determine it with cell voltage.

The reviewer is right that an accurate measurement of the anode and cathode potentials, along with presented data for cell voltage, would circle the data and complete the analysis related to non-standard copper refining process. However, although it appears that measurements of the separate potentials can be set simply, even with the quasi reference Cu wires, it could be hard to reach the reliability since the experiments are to be performed in a rather huge pilot industrial cell with the anodes and cathodes of corresponding huge geometry. Also, the distribution of temperature and electrolyte flow types through the cell volume, which can affect the uneven deposition on the cathode side could be the reason for the reliability measurements.  The higher reliability could be reached by separate measurements at different reference electrode positions with respect to the anode and cathode surfaces and then to perceive a 2D distribution of the potentials. However, these rather complicated measurements were out of scope of this manuscript, for which we decided to comment the only reliable data of the average potential difference comprised within measured cell voltage.

If there is data that measures the anode and cathode potentials separately, please show them together with the cell voltage.

No data

A more quantitative evaluation is possible if the theoretical dissolution and electrodeposition amounts are displayed together with the weight change of the anode and cathode in Table 2. In addition, current efficiency can be compared, so it will be more meaningful data. - 

Thanks for the comment. It is correct that the current efficiency is the important data for the electrorefining process but economic evaluation is out of scope of this manuscript. It is good suggestion for a new manuscript where will be discussed economic data of process for copper anodes with different impurities content.

Reviewer 2 Report

This paper is interesting but I think that far more information needs to be given on how the different chemical analyses were done, which techniques were applied to particular samples. I have several questions on the X-ray powder diffraction analyses. If these questions can be answered properly then I believe that this paper would be worthy of publication.

Page 4. Line 195. KJ solution is mentioned, what is this solution, it is not defined earlier in the text? Please clarify this point.

Page 5. Line 213. Please give more information on XRF. How were the samples prepared, what sort of instrument was used (EDXRF, WDXRF?) What was the anode of the XRF spectrometer.

Page 5. Line 226. Which techniques were used for determining chemical composition of anode, was this done by XRF or AES?

Page 9. Line 344. How were compositions of waste solutions analysed, please give more information?

Page 11. Line 408. Please give more information on how compositions were determined for anode slime.

Page 11. Line 440. How long did each XRD pattern measurement take?

Page 12. Line 442. Please give more info on this match software package.

Page 12. Line 446. Presumably this was the ICDD Powder Diffraction File database?

Page 12. Line 455. Figures 4ab and 6ab show two peaks around 8-9 degrees 2theta which are not assigned. Where do these peaks come from, are they due to the instrument? This information is needed to fully understand the XRD patterns.

Would it be possible to show the powder diffraction file pattern numbers for each phase detected by XRD?

Page 13. Line 489. Please offset these two patterns to make it easier to make a comparison. Figure 5 and Figure 7.

Page 14. Line 527. For future reference it is possible to estimate the % amorphous content by adding a known amount of an internal standard sample (e.g. silicon). From the difference between the real amount of silicon added and the amount of silicon determined by XRD the % amorphous can be estimated.

https://doi.org/10.1107/S0021889801002485

Author Response

Dear Reviewer,

thank you for you valuable comments and invested time! We are sending our answers to your questions (in bold letter)

This paper is interesting but I think that far more information needs to be given on how the different chemical analyses were done, which techniques were applied to particular samples. I have several questions on the X-ray powder diffraction analyses. If these questions can be answered properly then I believe that this paper would be worthy of publication.

We offered more information about different chemical analysis, and characterisation techniques!

The techniques that are applied to the particular samples are presented in part: 2.2.2. Characterization methods.

Page 4. Line 195. KJ solution is mentioned, what is this solution, it is not defined earlier in the text? Please clarify this point.

Added in the text:

2.1.4. Chemicals for preliminary testing the electrolyte mixing system

Pottasium iodide (purchased from Company: Merck KGaA, Frankfurter Str. 250, D-64271 DARMSTADT, Germany), was used for preparation 10 wt. % aqueous solution (KJ) that was used as electrolyte during preliminary testing of the system for electrolyte mixing.

Sodium thiosulphate (Na2S2O3.), purchased from Carl Roth GmbH + Co KG Schoemperlenstr. 3-5, D-76185 Karlsruhe, Germany) was used for preparation the aqueous solution for decolorization of KJ solution.

Preparation of the appropriate solutions was made with 18 MΩ cm deionized water.

Page 5. Line 213. Please give more information on XRF. How were the samples prepared, what sort of instrument was used (EDXRF, WDXRF?) What was the anode of the XRF spectrometer.

Added in our text:

Chemical analysis of copper anodes was done by Spector Xepos Energy Dispersive X-ray Fluorescence Spectroscope (ED-XRF, SPECTRO, Kleve, Germany). X-ray anode material is Au. Major and minor elements were determined via the fusion method (1000 â—¦C for 1 h with a mixture of Li2B4O7/KNO3 followed by direct dissolution in 10% HNO3 solution). Samples of copper anodes were obtained by cutting a part of the anode before polishing from the top, middle and the bottom  

Page 5. Line 226. Which techniques were used for determining chemical composition of anode, was this done by XRF or AES?

An XRF technique was used to determine the chemical composition of the anode. AES was used for analysis of metals in solution.

Page 9. Line 344. How were compositions of waste solutions analysed, please give more information?

Added in the text:

Each sample that analyzed at 24, 48 and 72 h was taken from the middle of the cells. Sludge precipitation is occurred in the samples that were analyzed without any previous preparation. Further, the samples were prepared by the next procedure: 10 ml of the electrolyte from cell medium, plus 10 ml of concentrated HCl and deionised water up to 50 ml. Appropriated dissolution was used for the higher concentration of the analyzed elements.

Page 11. Line 408. Please give more information on how compositions were determined for anode slime.

Added in the text:

Anode slime sample of 0.5 g was dissolved with aqua regia, transferred into volumetric flask of 50 ml and completed with deionized water. Appropriated dissolution was used for the higher concentration of the analyzed elements.

Page 11. Line 440. How long did each XRD pattern measurement take?

Added in the text:

Each sample measurement on the XRD apparatus lasted 40 min.

Page 12. Line 442.Please give more info on this match software package.

Added in the text:

Software package for XRD apparatus was COD database, COD REV44788 2012.03.20.

Page 12. Line 446. Presumably this was the ICDD Powder Diffraction File database?

Added in the text:

No, a COD database was used.

Page 12. Line 455. Figures 4ab and 6ab show two peaks around 8-9 degrees 2theta which are not assigned. Where do these peaks come from, are they due to the instrument? This information is needed to fully understand the XRD patterns.

Added in the text:

The peaks at 8-9 ° originate from the Compton effect and as such do not enter into the identification of the diffractogram. Since the recorded samples did not have characteristic peaks below 10 °, the revised diffractogram is shown in the range of 10 to 70 ° and this includes all specific peaks.

Would it be possible to show the powder diffraction file pattern numbers for each phase detected by XRD?

Yes, the corresponding card numbers from the COD database are given in Table 5.

Page 13. Line 489. Please offset these two patterns to make it easier to make a comparison. Figure 5 and Figure 7.

Peaks originating from the Compton effect at 8-9 ° were removed from the diffractogram by shifting the sampling range in the range of 10 to 70 °.

Page 14. Line 527. For future reference it is possible to estimate the % amorphous content by adding a known amount of an internal standard sample (e.g. silicon). From the difference between the real amount of silicon added and the amount of silicon determined by XRD the % amorphous can be estimated.

Thanks for this comments. It is included as idea in manuscript as a last sentence, part 3.4.1.